

# Mapping Water Content Dynamics in MAR-SAT systems using 3D Electrical Tomography

Lurdes Martinez-Landa[1,2,*], Jesús Carrera[2,3], Juan José Ledo[4], Perla Piña-Varas[5], Paola Sepúlveda-Ruiz[6], Montserrat Folch[6], Cristina Valhondo[2,3,*]

[1] Department of Civil and Environmental Engineering, Universitat Politècnica de Catalunya, Barcelona, Spain
[2] Associated Unit: Hydrogeology Group (UPC−CSIC)
[3] Geosciences department, Institute of Environmental Assessment and Water Research, Spanish Research Council (IDAEA-CSIC), Barcelona, Spain
[4] Department of Earth Physics and Astrophysics, Faculty of Physics, Universidad Complutense de Madrid, Spain
[5] Geomodels-UB Research Institute, Faculty of Earth Sciences, Universitat de Barcelona, Spain
[6] Biology, Sanitation and Environmental Department, University of Barcelona, Av. Joan XXIII, 08028 Barcelona, Spain

*Correspondence to*: cvalhondo@gmail.com

**Abstract.** The growing demand for high-quality water requires sustainable strategies to promote reuse and recycling. Managed
Aquifer Recharge systems, particularly Soil-Aquifer Treatment (SAT) systems, have demonstrated effectiveness in improving water quality by reducing contaminants through biodegradation, retention, and sorption. The coexistence of solid, liquid, and gas phases in the unsaturated zone (USZ) enhances adsorption, and retention of pathogens and colloids, while the availability of organic carbon and terminal electron acceptors sustains this zone as critical for biodegradation processes.

During recharge, the hydration-drainage front in the USZ follows a depth-dependent pattern. However, soil heterogeneity
causes water to infiltrate through preferential pathways, gradually hydrating the surrounding medium. This behavior is further influenced by system management (recharge strategy, applied flow rate, and/or installation of reactive barriers), which also promotes the development of biofilms. These biofilms facilitate water retention and act as localized microreactors for contaminant biodegradation.

We investigate the influence of recharge strategies (pulsed versus continuous) and the presence of a reactive barrier on the
hydration-drainage front in the USZ and biofilm development in two SAT systems. This was achieved using cross-hole electrical resistivity tomography and assessing biofilm formation through extracellular polymeric substances quantification in solid samples collected from the USZ during the recharge episodes.

The study compared two SAT systems: one consisting of fine sand and another with a reactive barrier incorporating of sand, woodchips, compost, biochar, zeolites, and clay. Two recharge episodes were analyzed: one with a continuous flow rate and
the other using a pulsed flow rate (while maintaining the same average flow rate across both episodes). Resistivity measurements, associated with the properties of the porous medium and the fluids circulating through it, were collected in the initial dry state and during the recharge, revealing the 3D distribution of the USZ volume during hydration (at the start of recharge) and drainage (when recharge ceased). Over time, these measurements also indicated the potential formation of





biofilms in the SAT system. Measurements at the beginning and end of each recharge period capture the 3D evolution of water
content.

Results showed that water infiltration occurred through preferential pathways or fingers, creating significant heterogeneity in
water content in both SAT systems. The reactive barrier enhanced water retention during dry periods, supporting biofilms
development. Furthermore, the pulsed recharge strategy promoted biofilm growth more effectively than the continuous
recharge strategy. These findings provide insight into optimizing recharge strategies and media composition to manage system
dynamics and, consequently, enhance contaminant removal in SAT systems.

## 1 Introduction

Groundwater depletion has become a pressing global issue driven by rising water demand and compounded by climate change
(Amanambu et al., 2020; Gurdak, 2017; Jasechko et al., 2024). Managed Aquifer Recharge (MAR) offers a robust and
sustainable solution to replenish aquifers while improving the quality of recharged water (Valhondo and Carrera, 2019). MAR
involves introducing water into aquifers via diverse methods to achieve objectives such as aquifer storage, water quality
enhancement, and prevention of seawater intrusion in coastal aquifers (Dillon, 2005; Page et al., 2018). However, regulatory
hurdles often constrain the widespread application of MAR (Alam et al., 2021). These regulations, aimed at ensuring water
quality due to global concerns about pollutants— such as endocrine disruptors, antibiotic-resistant bacteria, and nutrients—
focus heavily on the source water quality (Bijay-Singh and Craswell, 2021; Nnadozie and Odume, 2019; Thacharodi et al.,
2023). This focus often overlooks the natural pollutant reduction capacity of the porous media during recharge.

Pollutant removal is especially relevant for Soil Aquifer Treatment (SAT), a MAR technique that uses treated wastewater as
source for recharge. SAT systems have demonstrated remarkable water quality improvements, by removing essentially all
suspended solids and microorganisms, greatly reducing dissolved organic carbon, nitrate, phosphates and most metals (Fox et
al., 2001; Sanz et al., 2024a). To further optimize SAT, a reactive barrier has been proposed at the bottom of recharge basins
(Valhondo et al., 2020a). This barrier, comprising sand, woodchips, compost, biochar, zeolites, and clay, releases dissolved
organic carbon (DOC), fostering redox zonation and supporting diverse microbial communities. The reactive barrier also
enhances pollutant sorption, particularly for neutral and cationic compounds (Valhondo et al., 2023). Beyond its water
treatment benefits, the reactive barrier modifies the porous media texture, porosity, and water retention capacity (Martinez-
Landa et al., 2023). Studies have demonstrated its ability to reduce emerging contaminants, toxicity, and pathogen indicators,
making it a cost-effective, eco-friendly treatment technique for water reclamation and the recovery of aquifers (Sanz et al.,
2024c; Sunyer-Caldú et al., 2023). However, effective monitoring is essential for broader application and operation.

Monitoring SAT systems, especially in the unsaturated zone (USZ), presents unique challenges. While the aquifer can be
monitored via sampling wells and sensors, tracking dynamic processes in the USZ is more complex. Numerous studies
highlight that significant water quality improvements occur in the shallowest layers of soil, primarily driven by microbiological

processes (Bekele et al., 2011; Dillon et al., 2020; Fichtner et al., 2019). Microorganisms catalyze oxidation-reduction reactions, which consume organic compounds, including pollutants. In porous media, microorganisms often form biofilms as a survival strategy (Donlan and Costerton, 2002).

Biofilms are critical for water quality enhancement because biodegradation predominantly occurs within them. In fact, Jou-Claus et al. (2024) have shown that the most lipophilic organic compounds, including those that are toxic by accumulation,

tend to be retained in biofilms, which host the bacteria that degrade them, leading to high effective degradation rates even for recalcitrant compounds. Thus biofilms are essential for the optimal functioning of SAT systems. But biofilms alter the properties of the medium, affecting permeability and water retention (Baveye et al., 1998; Jou-Claus et al., 2024; Xia et al., 2014). The flow conditions in SAT systems (gravity flow, fine sand on top to prevent deep clogging and recharge rate much lower than the initial permeability) favor fingering, which further complicate monitoring. Point measurements, such as

moisture sensors or a quantification of bacterial activity in a sample, provide limited insight due to spatial variability and evolving flow geometries. Nevertheless, monitoring is crucial for effective system management.

Despite growing interest on SAT, visualization of water and microbial dynamics in the USZ remains elusive. Electrical resistivity tomography (ERT) offers a promising solution for monitoring the USZ in SAT systems. ERT is non-invasive, provides extensive monitoring coverage, and is well-suited to detecting changes in resistivity caused by water content, biofilm

growth, and preferential flow paths (Haaken et al., 2016; Wehrer et al., 2014; Wehrer and Slater, 2015). Previous studies have demonstrated ERT´s ability to identify areas of increased water content, preferential flow, and fingering in the USZ (Deiana et al., 2007; Koestel et al., 2008, 2009). When used in time-lapse mode, ERT can characterize flow dynamics over time, offering valuable insight into recharge processes (Nenna et al., 2014; Palacios et al., 2020). While ERT has been used to monitor aquifer during MAR, its application to study USZ processes in SAT systems remains unexplored (Sendrós et al.,

85    2021).

Our research addresses this gap by investigating the potential of ERT to identify preferential flow paths in two SAT systems: one with reactive barrier, which is hypothesized to promote biofilm development and enhance water retention, and another without reactive barrier. We utilize ERT to monitor the temporal evolution of these two SAT systems during two recharge episodes: one with continuous inflow and the other with pulsed inflow. We then link resistivity variations to flow patterns and

biofilm dynamics within the USZ.

## 2. Materials and Methods

### 2.1. SAT systems

ERT acquisitions were conducted at the pilot SAT systems located within the Palamós Wastewater Treatment Plant (WWTP) in NE Spain. For a more detailed description of the WWTP and local weather conditions, see Section SI.1 and Figure SI.1 in



the Supplementary Information. These systems, described in detail by Valhondo et al. (2020b), consist of six SAT replicas. Each replica represents a 3.5 m² surface recharge basin overlaying a 1.15 m thick USZ. Beneath this a 15 m-long x 2.34 m-wide canal filled with well-graded fine sand (grain size ~ 0.4 mm) that simulates the aquifer (Figure 1).

A 1 m thick Reactive Barrier (RB) was installed between the basin surface and the aquifer to promote pollutant removal through adsorption and biodegradation. The RB design includes a 20 cm fine sand layer on top to prevent deep clogging and
floating of woody materials and a 20 cm coarse sand layer below to evenly distribute recharged water. This study focusses on two specific systems: Sand System (SandS), which consists solely of fine sand, thus representing traditional SAT systems and serving as reference, and Reactive Barrier System (RBS), composed of 50% sand, 30% organic matter (a blend of wood and garden wood compost), 10%Biochar, 8%zeolite, 2%clay. The RB enhances water retention due to the secondary porosity of the pieces of wood, biochar and zeolite. Retention curves indicate the porosity of RB is some 50% larger than that of the Sand
(0.59 vs. 0.38, respectively) (Figure SI.2).

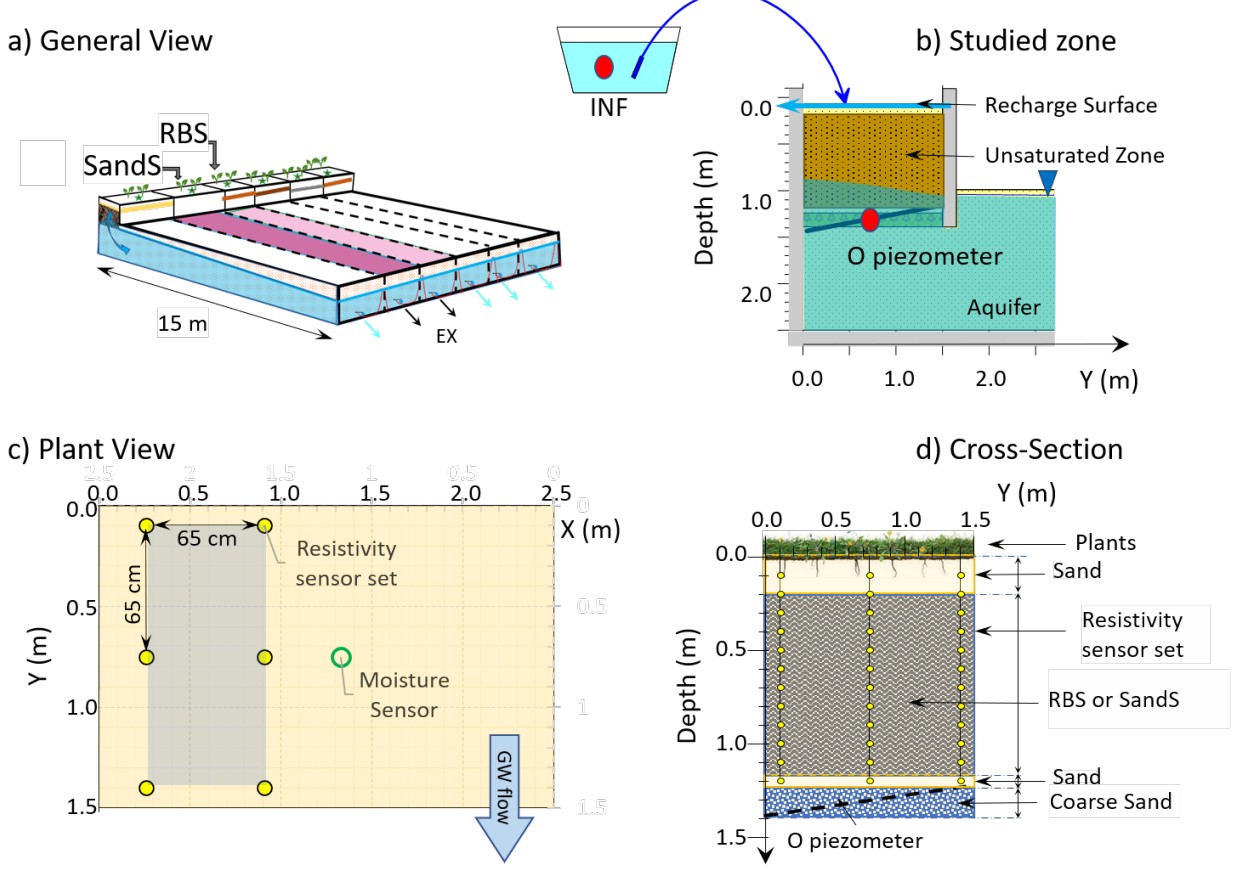

**Figure 1: Site description: a) General view of the six SAT systems in Palamós, including the two studied systems, SandS and RBS; b) Longitudinal cross section the SATs systems; c) Plan view; and d) Lateral cross section showing the location of ERT electrodes.**



Vegetation on the recharge basin plays a critical role in system functionality by reducing clogging and potentially enhancing pollutant removal (Valhondo et al., 2020b). Grass was planted following the installation of these RBs in January 2021. The grass was established using pre-grown sods. The soil within the sod roots zone consisted of a mixture of sand and small proportion of clay. In some systems, the clay content was slightly higher, initially reducing infiltration capacity. This issue was mitigated by perforating the sods layer for several weeks during the first RP, following the recommendations of a soil

scientist. As vegetation progressed, the problem was resolved naturally. No further maintenance was performed, allowing vegetation to evolve freely, although it remained sensitive to seasonal changes and recharge cycles (i.e., dry or flooded periods). For further details on the SAT systems see Figure SI.3 in the Supporting Information.

The SAT systems are fed with secondary effluent from the WWTP (hereafter referred to as "INF"), which is pumped from a tank located at the outlet using dosing pumps (PRIUS, Emec s.r.l., Rieti, Italy). Pumping rates are recorded continuously with

electromagnetic ISOMAG MS600 flowmeters (ISOIL industria, Milan, Italy). The pumped volume is distributed over the recharge surface using drip irrigation pipes (their layout can be seen in Figure SI.3). Discharge occurs from the base of the aquifer, through a level control outlet at the opposite end of the recharge basins, EX point (at the right of Figure 1-a). The discharge point of the systems is equipped with a water meter to record the discharge volume.

Each SAT system includes 10 piezometers equipped with CTDs sensors (DIVER, VanEssen Instruments, Waterloo, ON,

Canada) that record aquifer heads, temperatures and Electrical Conductivity (EC) in the saturated points (Figure 1-b). Temperature and volumetric water content were measured using 6 capacitive moisture Aquacheck-sensors (Cesens, Logroño, La Rioja, Spain) equally spaced 10 cm at 6 depths in the central recharge basin area (Figure 1-c). Weather data, including rainfall, wind direction and velocity, atmospheric moisture, and temperature, was recorded onsite by a weather station (Meters) (Figure-SI.1).

For the ERT acquisitions, six rods (spaced 0.65 m apart) were installed, each equipped with 12 electrodes. The electrodes spaced 10 cm apart and extending from the basin surface to the base of the USZ (depths range: 0.1 to 1.2 m), encompass a monitored volume of approximately 1 m³ (Figure 1-c and 1-d). To maintain consistent ERT measurements, this volume remained undisturbed and free of metallic objects, which could interfere with readings. Half of the USZ volume was allocated for ERT acquisitions, while the remaining half accommodated additional instruments and material sampling for biofilm growth

studies over time.

## 2.2. Recharge systems operation

The SAT systems operate by alternating between Wet Cycles (WC), during which recharge takes place, and Dry Cycles (DC), during which the recharge is paused to allow the USZ to drain and the surface to dry. This alternation is necessary to restore infiltration capacity, which can decline due to clogging caused by biological activity (e.g., biofilm growth) and the





accumulation of fine organic and inorganic particles on the basin surface. A common guideline recommends alternating two weeks of WC with two weeks of DC (Bouwer, 2002). However, the optimal schedule depends on site-specific conditions and the intended treatment objectives of the SAT system, as different contaminants may require distinct hydrological regimes to maximize their removal efficiency. Our observations indicate that our SAT systems have operated continuously for periods of several months without clogging (having to stop for maintenance), which we attribute to the role of vegetation in maintaining

open recharge pathways. In this study, we implemented extended WC durations to evaluate two distinct recharge strategies: continuous (Cont) and pulsed (Puls) operations. In the Continuous strategy, recharge was applied at a constant flow rate, promoting steady-state flow conditions. In the Puls strategy, recharge was applied for 12 hours per day. This approach (Puls) enhances oxygen availability in the USZ, as air is sucked into the USZ during DCs and pushed down during WCs (Roumelis et al., 2025; Sepulveda-Ruiz et al., 2025). Under Puls operation, the recharge flow rate was doubled compared to Cont

operations to ensure identical daily recharge volumes (Figure 2-a and Figure SI.4a). Two recharge episodes were monitored in this study:

    1-  Pulsed operation: The Wet Cycle started on September 21, 2022 (StrPuls), after 110 d of Dry Cycle, and ended on November 11, 2022 (EndPuls),

    2-  Continuous operation: The Wet Cycle started on December 22, 2022 (StrCont), after 41 d of Dry Cycle, and
concluded on March 22, 2023 (EndCont).

### 2.4. Cross-hole electrical resistivity tomography

Cross-borehole ERT involves measuring the ratio of the voltage between an electrode pair to the current injected between another electrode pair, across numerous sets of four electrodes positioned along the considered boreholes. For this experiment 6 plastic rods (borehole) with 12 electrodes spaced at 10 cm intervals on each array. The uppermost electrode on each array

was 10 cm below the tank top (see Figure 1-c). Bellmunt et al. (2016) suggest that it is better to use different configurations (dipole–dipole, pole–tripole and Wenner) with different sensitivity patterns to obtain the maximum information about the subsurface resistivity. The configurations adopted here were the ones described and assessed by Bing and Greenhalgh, (2000) and Bellmunt et al. (2016).

The measurements were conducted using a Syscal Pro multi-channel (10-channel) system (IRIS instruments) with 72
electrodes. Current injection lasted $250 \cdot 10^{-3}$ s, with up to six stacks performed to ensure data quality. Each tomography session lasted over 80 minutes and 6740 measurements were acquired. Measurements were timed to occur just before cycle transitions (e.g., End Wet Cycle: EndWC or End Dry Cycle: EndDC) and again four hours after the new cycle began (Start Wet Cycle: StrWC or Start Dry Cycle: StrDC). These intervals were chosen to capture the evolution of water distribution within the USZ.

The numerical inversion was performed using the R3t code via the ResIPy open-source package (Binley, 2015; Binley and
Kemna, 2005; Blanchy et al., 2020). R3t code provides a forward/inverse solution for three-dimensional current flow within a





tetrahedral mesh. The inverse solution is derived from a regularized objective function combined with weighted least squares, in an Occam-type approach. We follow the strategy proposed by Bellmunt and Marcuello (2011) for the quality control of the data based on the comparison between normal and reciprocal measurements. We chose a threshold of 15 % difference between the normal and reciprocal data to keep the measurement. Moreover, during the inversion process a linear error model relating
the reciprocal error with the measured resistance is used to improve the inversion results and to reduce the variance in the model estimates following (Tso et al., 2017).

### 2.3. Biofilm monitoring in the USZ via EPS extraction

Samples from the USZ material were collected weekly during both the Pulsed and Continuous recharge periods to evaluate biofilm growth via quantification of extracellular polymeric substances (EPS). Sampling began at the start of each wet cycle
(i.e., StrWC) and continued until the final day, in both Sand and RB systems. Using a soil probe, samples were taken at three depths (28, 41 and 54 cm), outside the ERT monitoring area and ensuring that previous sampling points were avoided. Samples were stored in sterilized plastic tubes, transported at 4 °C, and preserved at -20 °C until analysis. The holes were refilled with the same type of barrier material that had sampled.

EPS are key indicators of biofilm development. They are primarily composed of polysaccharides and proteins and contributes
more to bioclogging than microbial cells themselves (Mitchell and Nevo, 1964). EPS were extracted using a cation exchange resin (CER) (Dowex Marathon C, Na+ form, strongly acidic, Sigma-Aldrich, Steinheim, Germany), which was pre-conditioned according to the manufacturer's instructions.

For each sample, 5 g of material was transferred to a falcon tube containing 30 ml of phosphate buffer (1.78 g of $Na_2HPO_4 \cdot 2H_2O$, 0.24 g of $KH_2PO_4$, 8 g of NaCl, 0.2 g of KCl; pH 7.0) and 5 g of preconditioned resin. The EPS extraction was conducted
at 4 °C on a shaker (300 rpm) for one hour to minimize microbial cell disruption. Following this, the samples were centrifuged at 12,000×g for 15 minutes at 4 °C (Eppendorf 5810R, Eppendorf Ibérica S.L.U., Spain). The supernatant was collected and stored at 4 °C for subsequent EPS characterization, with resin concentration optimized to maximize EPS recovery (Frølund et al., 1996). Polysaccharide content, as a component of EPS, was quantified using the phenol-sulphuric acid assay (DuBois et al., 1956). Standard curves for glucose (0 – 200 µg/ml) and BSA (0 – 25 µg/ml) were prepared.

Polysaccharide content was measured following the phenol-sulphuric acid assay and protein content was measured following the Bradford assay. Standard of glucose (0 – 200 µg/ml) and BSA (0 – 25 µg/ml) were prepared (DuBois et al., 1956). The results are presented in total EPS content (µg/g dw) (polysaccharides and proteins) during both recharge episodes (See table SI. 1).

### 3. Results and Discussion

*3.1- Influence of the recharge strategy on hydraulic conditions:*



The hydraulic evolution of RBS and SandS are shown in Figure 2 and 3, and the complete evolution of point moisture measurements with depth for both systems are displayed in Figure-SI. 4 and SI. 5 in the Supporting Information. Recharge rate peaks ($q_{rec}$) during Puls are double than during Cont (Figure 2 and 3, -a), which is reflected in the evolution of volumetric water content (Figure 2 and 3, -b). During the first four days of Puls, a malfunction in the automatic pump controller caused
continuous recharge instead of pulses, leading to a step increase in %H. After this, Puls WC proceeded as intended, with normal recharge rates ($q_{rec}$) and corresponding system responses in %H, Tª, and aquifer head (depth to water). The electrical conductivity (EC) measured in the INF water averaged 2 mS/cm, with variations associated with specific wind directions during certain rainfall events that caused seawater intrusions into the sewerage system as a result of wave overtopping. (see figure SI. 6). Similar variations were subsequently detected in the O piezometer, but with lower and slightly broader peaks,
allowing EC to be used as a tracer for estimating the residence time of the recharged water within the USZ (Martinez-Landa et al., 2023; Valhondo et al., 2016). The Aquacheck probe includes temperature sensors placed at the same depths as the moisture sensors. Figure SI. 7 illustrates the temperature variations recorded at 0.4 m depth for both systems and system operations.

At the start of the Puls period (blue dotted line), both systems were extremely dry due to the preceding 110-day dry cycle
(EndDC). Lower moisture levels may enhance the formation of preferential flow paths (Kang et al., 2023). Water content was below 5% in SandS, while RBS showed slightly higher values (around 10%). By the end of the wet cycle (EndWC), water content had increased to 20-25% in the SandS and to 30-35% in the RBS, highlighting the greater moisture retention capacity of the materials integrated in the RB compared to sand. Variations in %H due to Puls recharge reached around 15% in SandS and 20% in RBS at 10 cm depth. These fluctuations persisted with depth in SandS but became smoother in the RBS, with
water content stabilizing at ~ 20% in SandS and ~ 35% in the RBS (Figures SI. 4 and SI. 5 respectively).

Under Cont recharge (red line), both systems started with higher initial water contents than in Puls, but SandS exhibited lower values (~10%) compared to RBS (~15-30%, increasing with depth). At the end of the Cont WC (EndWC), water content reached ~20% in SandS and ~30% in RBS, with minimal variation with depth. By the End of the Cont scheme, we have observed the development of a water layer due to the reduction of infiltration capacity. In these types of systems, the reduction
in infiltration capacity is likely attributable to: (i) the superficial accumulation of fine organic particulate material transported by the INF water; (ii) increased water viscosity associated with lower winter temperatures (Figure SI.7); (iii) seasonal stages in the life cycle of certain plant species, during which established plants cease proliferating and produce fewer new roots (Figure SI.3; Xiao et al., 2024); and (iv) a slightly higher proportion of clay in the root zone (observed in SandS).. These conditions favored the temporary formation of a surface water layer (Figure SI. 10, and SI. 11).

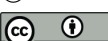


**Figure 2: Time (days after the beginning of each WC) evolution of main hydraulic variables of the SandS: a) recharge rate, b) volumetric water content at 40 cm depth (other depts displayed in the SI), and c) materials (color shadows) and water level at piezometer O, below the barrier, during both recharge cycles (Puls in blue, and Cont in red). The left and right columns are zooms at the initial and final times to best display the times of ERT acquisitions (dashed lines). Note that the recharge periods (RP) were sequential. After a long dry period (EndDC), pulse recharge started at StrWC (=StrPuls) and continued for 51 days until EndWC(=EndPuls=StrDC). The SAT remained dry for 41 days until a continuous RP started at StrWC (=StrCont), which remained operational for 91 days until EndWC(=EndCont). The time scale around this last stop has been shifted to facilitate superposition with EndPuls.**

Notably, in SandS, dissolved oxygen at 35 cm depth remained above 3 mg/L for most of the Cont WC but dropped sharply around day 50 to negligible values (Figure SI. 8 and SI. 9), coinciding with the period just before surface water began to accumulate and with the peak EPS concentrations recorded in SandS (Figure 4). The water content did not decrease with depth





during this period, indicating that this was not a case of complete surface clogging which would have reduced USZ water content once drainage exceeded infiltration.

Depth to water in RBs (Figure 2 and 3, -c) decreases in response to recharge from initially dry (1.2 m) and fluctuates during pulsed recharge. Considering that the aquifer water tables are measured in terms of depth, the water table elevation is observed as a decrease in depth.

Figure 3: Time (days after the beginning of each WC) evolution of main hydraulic variables of the RBS: a) recharge rate, b) volumetric water content at 40 cm depth (other depts displayed in the SI), and c) materials (color shadows) and water level at piezometer O, below the barrier, during both recharge cycles (Puls in blue, and Cont in red). The left and right columns are zooms at the initial and final times to best display the times of ERT acquisitions (dashed lines).



Since Puls and Cont WCs had different durations, data analysis focused on the first 51 days to observe the systems' response to the USZ saturation behavior (Figure 2 and 3, ). Cont WC continued until day 91, exhibiting trends like those of Puls WC.

Wi-Fi connectivity issues with the Aquacheck datalogger caused some data loss during Cont measurements. However, once the signal was restored, the measurements trends aligned with prior observations. The CTD sensor recorded data only after recharge water reached its location, resulting in initial gaps in WC measurements.

*3.2- Influence of the recharge strategy on EPS:*

EPS content differed markedly between the two SAT systems (Figure 4). In SandS, average EPS concentrations were 71 µg/g during Puls recharge and 69 µg/g during Cont recharge. In contrast, RBS exhibited significantly higher EPS levels, averaging 390 µg/g and 225 µg/g during Puls and Cont recharge, respectively (Table SI. 1).

In SandS, EPS levels began to rise on day 22 (Puls) and day 25 (Cont), displaying similar temporal trends across both strategies.
In RBS, a sharp increase in EPS was observed by day 15 during Puls recharge, followed by a decline, whereas under Cont recharge, EPS levels increased gradually from day 20 to day 34. This Puls sample on day 15, may reflect the heterogeneity of the medium, as it does not exhibit a consistent increasing trend but rather a localized increase.

EPS production typically occurs during the initial phase of microbial attachment to surfaces and plays a key role modifying porous media properties, including pore-scale hydration (Du et al., 2021; Lu et al., 2024). The earlier peak observed during
the Puls strategy may suggest that it accelerates the microbial attachment phase. However, this effect could also be influenced by higher temperatures during the Puls recharge period compared to the Cont period (see Figure SI.7 for temperature evolution).

The differences between the two SAT systems are likely related to the organic-rich composition of the RB which promote both water retention and microbial colonization. Sustained moisture levels (%H) coupled with elevated temperature, further support
biofilm growth. In turn, the biofilm enhances moisture retention, creating a positive feedback loop that further supports biofilm persistence and system functionality.

This interplay is relevant in SAT systems, as biofilm presence may facilitate the degradation of contaminants of emerging concern, while also potentially enhancing the transport of microbial contaminants (Du et al., 2021; Jou-Claus et al., 2024).


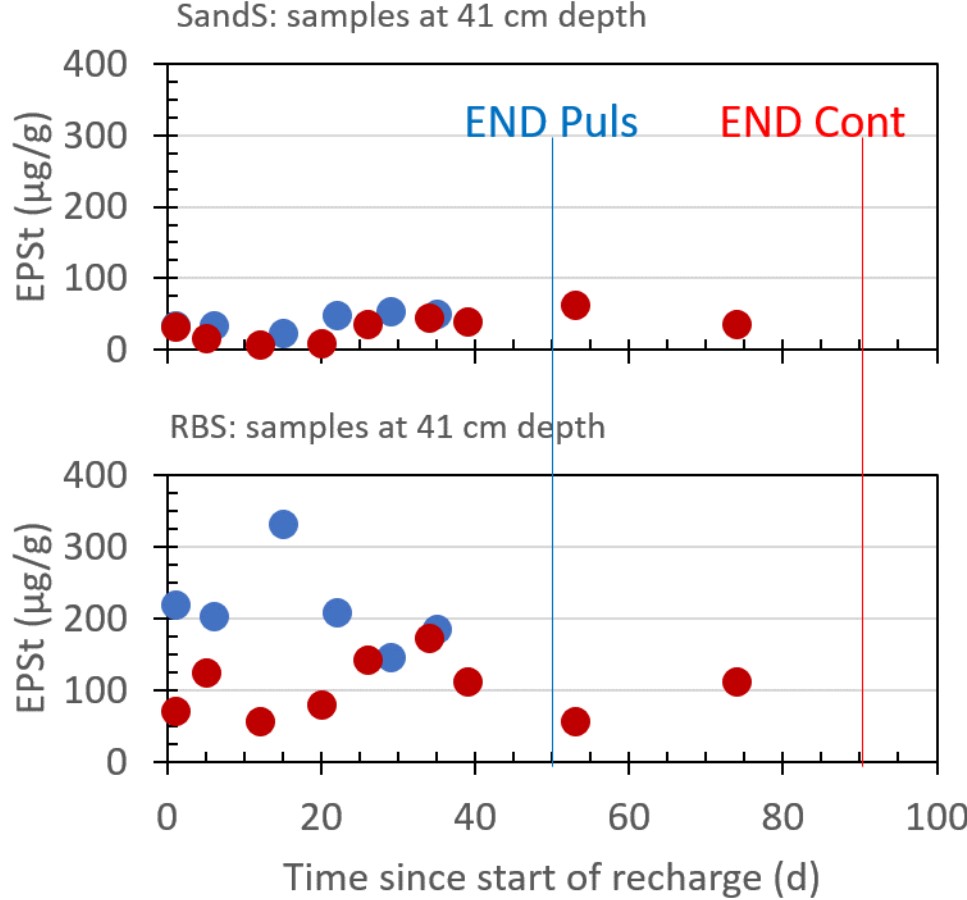

**Figure 4: Total EPS (µg/g) sampled in the USZ material at 41 cm depth at the SandS system (up) and RBS system (down). Blue and red dots correspond to values obtained during the pulsed (Puls) and the continuous (Cont) recharge strategies, respectively. Due to different durations of each Wet Cycle (51 d for Puls and 91 d for Cont), fewer samples were taken in Puls than in Cont RP.**

*3.3- Spatial Moisture Dynamics Revealed by ERT:*

The ERT measurements provide spatially resolved insight into moisture distribution across the USZ (1 m³ block), from the basin surface to the aquifer water level, complementing the point-scale data discussed in previous sections. Results are displayed in three perpendicular planes, as shown in Figures 5, 6 and 7. These figures depict a temporal sequence of specific measurements taken with the ERT. Log10 resistivity values are inversely related to water content (%H). Dark blue areas correspond to the lowest resistivity values, indicating the wettest (i.e., highest %H), while light blue areas represent the driest (lowest %H values). The CTD sensor automatically recorded water EC every 30 minutes at the INF (WWTP effluent tank)





and the O piezometer (beneath the USZ) (Figure 1), with an average value of 2.2 mS/cm. The Lowest water EC values occurred

during heavy rain events, while peak values were linked to seawater intrusion into the sewage system caused by coastal windstorms (Figure SI.6). ERT results are presented in terms of inverted ERT to facilitate consistent comparison within acquisitions (all water displays the same EC) but slightly hinder long term comparisons (note that water EC fluctuations are far smaller than those of ER).

The sequence starts with the Puls RP during the driest period (EndDC), where light colors predominate. Infiltration is visible

in the second frame (StrWC= StrPuls), where it becomes evident that it does not occur as a homogeneous front, but rather in preferential areas, resulting in darker spots within the lighter regions. Infiltration appears to occur more easily in SandS than in RBS, as indicated by the increasing dark blue areas with depth in the second frame for this system. This observation is consistent with the lower initial moisture content in the SandS, which may facilitate preferential flow paths and has got less water content, and resident water to displace (compare Figure 2-b for Sand and 3-b for RBS).

At the EndPuls dry areas are still present amid more humid zones, showing heterogeneity in water content distribution measurement, despite the high %H. Water content is higher in RBS than in SandS, which aligns with the higher EPS concentrations observed. This is consistent with feedback mechanism, where higher moisture supports EPS maintenance, and EPS in turn enhances water retention. The fourth frame (StrDC) shows the first hours of drainage, revealing again greater retention capacity and heterogeneity in RBS, reflected by a broader range of color gradation, which is again consistent with

the organic-rich materials and high biofilm content that enhance moisture retention as shown in Figure SI. 2.

Cont began after 41 days of continuous drainage in winter (EndDC), which represents the first ERT measurement in this sequence. SandS is drier than RBS, as expected. Four hours after the recharge begins, the StrWC frame shows increase %H in both systems, with RBS displaying a more distinct preferential flow path than SandS.

After 91 days of continuous recharge, the EndWC measurement is taken. In this sequence, it appears that more preferential

flow paths have developed compared to Puls, as evidenced by the increased heterogeneity in color distribution across the three figures. By the end of the continuous recharge phase (EndWC), water accumulation is observed at the surface of the basin, 28 cm in SandS and 1.5 cm in RBS. During the 4 hours leading to StrDC, the SandS water layer decreases by 6 cm, while RBS water layer disappears. The levels in the aquifer (Depth to Water: DW), indicated with a red line in the figures, slightly saturate the lower part of the analyzed volume at the end of the two WC (EndWC) measurements, with %H increasing due to capillary

rise.



**Figure 5: X depth Cross section at 60 cm in Y direction, highlighted in red in the lower-right scheme. The top timeline displays the
measurement periods: the recharge period (RP: Pulse or Continuous) and their start or end points; the central timeline indicates
the cycles (Wet Cycle, WC, or Dry Cycle, DC) within each RP and their durations in days; and the bottom timeline shows time in
minutes, referenced to the start or end of each cycle when the ERT data was acquired (red for dry cycles, blue for wet cycles). The
central section of the figure contains the resistivity measurements, arranged in two rows: the top row shows measurements for the
SandS and the bottom row shows measurements for the RBS. The legend indicates the resistivity scale (log10) and its inverse
correlation with water content (%). In the downer part of each image the Deep to Water (DW) is indicated in red, to mark the
position of the aquifer (100% of water content). A blue square in the upper part of the images indicates the water layer over the
basin surface, if exist, and the numbers the sizes in cm at the time of ERT measurement.**



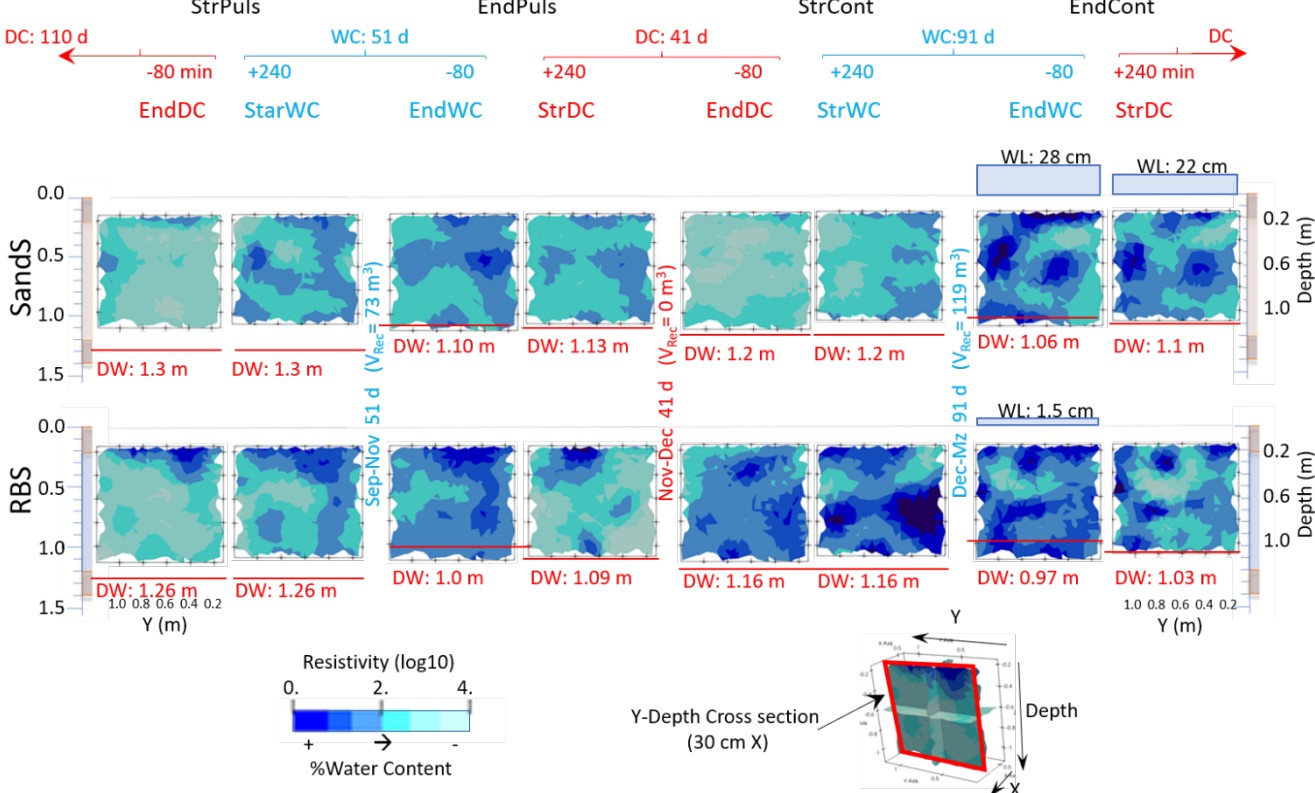

**Figure 6: Y depth Cross section at 30 cm in X direction, highlighted in red in the lower-right scheme. The top timeline displays the measurement periods: the recharge period (RP: Pulse or Continuous) and their start or end points; the central timeline indicates the cycles (Wet Cycle, WC, or Dry Cycle, DC) within each RP and their durations in days; and the bottom timeline shows time in minutes, referenced to the start or end of each cycle when the ERT data was acquired (red for dry cycles, blue for wet cycles). The central section of the figure contains the resistivity measurements, arranged in two rows: the top row shows measurements for the SandS, and the bottom row shows measurements for the RBS. The legend indicates the resistivity scale (log10) and its inverse correlation with water content (%). In the downer part of each image the Deep to Water (DW) is indicated in red, to mark the position of the aquifer (100% of water content). A blue square in the upper part of the images indicates the water layer over the basin surface, if exist, and the numbers the sizes in cm at the time of ERT measurement.**



Figure 7: XY horizontal section at a depth of 60 cm, highlighted in red in the lower-right scheme. The top timeline displays the
measurement periods: the recharge period (RP: Pulse or Continuous) and their start or end points; the central timeline indicates
the cycles (Wet Cycle, WC, or Dry Cycle, DC) within each RP and their durations in days; and the bottom timeline shows time in
minutes, referenced to the start or end of each cycle when the ERT data was acquired (red for dry cycles, blue for wet cycles). The
central section of the figure contains the resistivity measurements, arranged in two rows: the top row shows measurements for the
SandS, and the bottom row shows measurements for the RBS. The legend indicates the resistivity scale (log10) and its inverse
correlation with water content (%).



Analyzing the validity of the inversion is required prior to a more detailed analysis of results. The first validation comes from consistency with expectations, as discussed in the two previous paragraphs. But a more precise discussion of the validity comes from comparing inversions at the end of wet cycles (EndWC) to those at the beginning of dry cycles (StrDC). What happens

in between is the continuation of recharge for 80 min, which should imply little change with respect to EndWC inversions, followed by 240 minutes draining the retained water. This situation occurred twice (at EndPuls and at EndCont) for each system (SandS and RBS). Comparing the EndWC to the StrDC (columns 3-4 of Figure 5through 7 for EndPuls, and 7-8 for EndCont). These comparisons yield several useful messages. But the first one is that all of them reflect with surprising detail the draining process. This is hard to visualize because it occurs in 3D and one needs to examine Figures 5 through 7, while

simultaneously imagining gravity drainage, to properly understand the evolution. Therefore, we only discuss in detail the evolution of the RBS during the EndPuls (bottom row maps in columns 3-4 of the three figures). A finger can be seen at the end of the WC (column 3) in Figure 6 at $Y \approx 0.3\ m$ extending from the surface to nearly the bottom, where it spreads laterally. This finger extends over the whole width at the horizontal cross section (Figure 7) but is not quite crossed by the vertical cross section of Figure 5(only the edges). It would be expected that this finger would serve as drain of the USZ when recharge stops.

This drainage becomes apparent when examining the StrDC (column 4). Areas away from the finger stay wet. Also relevant is the significant reduction of moisture at the bottom, which coincides with the coarse sand base. Similar observations can be made in the other three EndWC-StrDC pairs. It is also interesting the behavior at EndCont where both systems were flooded at EndCont, partly due to the low temperatures (Figures SI.1 and SI.7). But SandS remained flooded at StrDC, while RBS had become dry (the thickness of the water layer flooding is shown in Figure 5 and 6, and its time evolution in Figures SI. 10a and

SI. 11a). In this case the resistivity maps at EndWC and StrDC are very similar for SandS (top row maps in columns 7-8 of the three figures) but suggest beginning of drainage for RBS. Considering that all inversions were performed independently from independent ERT acquisitions, we take this consistency to indicate that the resistivity variability displayed in Figure 5through 7 is not an inversion artifact, but representative of the SAT system behavior, which allows us to discuss other features of these maps.

The EndWC-StrDC pairs, discussed in the previous paragraph, also provide information about biofilms and the conditions for growth and survival. Dark blue spots can be observed at the end of the recharge cycles (the 4 EndWC maps in columns 3 and 7 of Figure 5through 7 ) remain slightly reduced in size, after 4 hours of drainage. Given that the permeability of the systems is of the order of 10 m/d, and that the raw media are relatively homogeneous, they should have drained if they had represented simply water saturation. We conclude that they must represent areas with significant water accumulation due to high organic

content and biofilms. In short, very low resistivity area (dark blue in the figures) can be associated with high biofilm content.

Large biofilm clusters do not occupy the whole medium, but a relatively small fraction, as can be deduced from the result of the previous paragraph plus close examination of the Figures. As expected, the remaining wet immediately after drainage starts (again, EndWC-StrDC pairs) given their low effective permeability. The relatively low biofilm filled area might suggest that only a small fraction of the USZ volume is active. ERT resolution does not allow us to reach such conclusion. Biofilm clusters



may have a broad range of sizes. Kurz et al. (2023) show that they follow a power law distribution, which was required by Wang et al. (2024) to fit breakthrough curves of partitioning tracers in biofilm growth laboratory experiments. Therefore, one may expect significant biochemical activity in biofilm clusters not visible in the ERT inversion. Still, one may also expect dramatically reducing conditions, and thus different microbial communities, in the inner cores of the large clusters, which should be broadened the range of contaminants removed during SAT.

Significant differences could be observed on the behavior of the SandS and RBS during the dry period, as can be deduced from comparison of the 4 and 5 columns of StrDC-EndDC pairs in Figure 5through 7. It is clear that the SandS drains well, which favors aeration, oxygen diffusion into the USZ, and the consumption of EPS on biofilms. In fact, it is this type of response what motivates traditional operation of SAT systems since Bouwer (1974). To prevent biological clogging, the usual practice is to alternate wet and dry periods, typically with durations around 2 weeks each. Infiltration rates usually recover after two

weeks dry which conventional wisdom attributes to biofilm consumption under aerobic conditions. This is consistent with what we observed in the SandS. The RBS showed much larger moisture at EndDC than at StrDC. While this increase can be attributed to the rainfall occurring at the end of DC, the fact remains that the SandS was dry at EndDC and the RBS was not. During the Puls scheme, the SandS maintains aerobic conditions throughout the entire USZ, whereas the RBS maintains aerobic conditions only during the dry part of each pulse, and only in the upper part of the USZ, sustaining anaerobic conditions

at depth, which are reversed at the beginning of the dry cycle (StrDC), see oxygen behavior in Figures SI. 8 (SandS) and SI. 9 (RBS). We attribute this differing behavior to the high retention capacity of both the organic matter and the biofilms in the RBS.

    Figure 5through 7 allow visualizing also differences in the behavior of the SandS and RBS during the two recharge cycles. The SandS consistently starts with relatively low water retention during the first hours, but it increases significantly both during

Puls and Cont recharge regimes. We attribute this increase to biofilm growth. Similar behavior occurs for the RBS during the Puls cycle, but not during the Cont cycle. In fact, as discussed in the previous paragraph, the RBS had large water contents at EndWC, so it is not surprising that it immediately showed very high moisture at StrWC (column 6, bottom row Figure 5through 7). What is surprising is that water retention, while still higher than the SandS, decreased by the next acquisition (EndWC, column 7). We attribute this behavior to a reduction in infiltration capacity driven by the combined effect of several factors as

it is explained in the 3.1 section, which may have decreased biological activity. Notably, this clogging effect was not observed during a six-month Cont RP conducted before the another sod installation (Sanz et al., 2024c, b). Therefore, we deduce that, in this case, the slightly higher proportion of clay in the root zone of the new sods, together with the accumulation of fine organic particles and the long duration of Cont recharge scheme, is responsible for the observed reduction in infiltration capacity.

The evolution of oxygen concentrations in both systems during both cycles is also interesting (Figures SI.8 and SI.9). As expected, oxygen concentrations remain high during the pulsed cycles (recall, section 2.2, that the goal of pulses was to enhance



oxygen availability in the USZ). However, concentrations in the SandS remained much higher (fluctuating around 6 mg/L at 35 cm depth and 4-5 mg/L at 90 cm depth, close to the water table) than in the RBS (concentrations dropped after 30 days of recharge and started fluctuating between below 0.01, LOD, and 2 mg/L at 35 cm, and between 0.03 and 0.1 mg/L at 90 cm).

This reflects two factors. On the one hand, SandS showed faster drainage than RBS, which favors oxygen diffusion. On the other hand, the lower concentrations in the RBS reflect the biofilm (very low resistivity) virtually covering the top of the RBS during Puls (column 3, bottom row of Figures 5 and 6). The fact that oxygen was virtually depleted at 35 cm during the Puls WC suggests a very high level of biological activity. Perhaps more significant is that Oxygen never reached air saturation, even though it is clear that it was being sucked (presumably across preferential air paths, in a process similar to the "aquifer

breathing reported by Roumelis et al. (2025), except that in their case it was driven by aquifer head fluctuations, while ours was driven by drainage fluctuations. A last comment regarding oxygen concentrations is that recorded values, during Cont recharge, were higher at 90 cm depth than at 35 cm. This result suggests that oxygen rich water was reaching the deep portions of the USZ through preferential flow paths, not recorded by our sensors. This illustrates one of the advantages of ERT overall maps with respect to point measurements.

Figure 5through 7 also show that the "hot spots" (dark blue areas indicative of big biofilm clusters) changed location along the recharge cycles. We interpret this displacement as indicative of biochemical processes. Big clusters may grow in the edges where electron donors and acceptors flow close to large microbial communities but may die in the middle for lack of nutrients. A hint of this process can be inferred by comparing the horizontal cross sections of the RBS before and after of EndPuls (Columns 3 and 4, respectively, of the bottom row of Figure 7). The apparent finger intersected by this section at around Y=0.4

before EndPuls, appears displaced towards Y=0.5 (upwards in the figure) immediately after the stop of injection, which suggests that the biofilm is growing in that direction, so that it is where water is retained at StrDC. Intermittency in preferential flow paths has also been observed by Kurz et al. (2022) using microfluidics techniques. They attribute it to the competition between growth, similar patterns as those described here, and shear stress that causes rapid opening of preferential flow paths. Their head gradients ranged between 80 and 300, compared to close to 1 in our experiment, so that it is not clear if their

conclusion can be extrapolated to field conditions.

## 4. Conclusions

The first observation of this work is that ERT allows visualizing and monitoring variations in water retention that we associate to biofilm growth. This is relevant because contaminant removal in SAT systems is driven by sorption and biodegradation processes. It is known that biodegradation occurs predominantly within biofilms, both because they host the bacterial colonies

that promote degradation and because they tend to sorb the most lipophilic compounds by mechanisms analogous to those that favor their accumulation in animal tissues. The implication is that, on the one hand, it is desirable to promote biofilm growth.



But, on the other hand, biofilms tend to clog. Navigating these contradicting requirements will require detailed monitoring. Our first conclusion is that ERT will help with such monitoring.

Beyond its potential use for detailed monitoring, ERT's ability to map water distribution heterogeneity has proven useful in gaining insight on the functioning of different recharge strategies and the use of a reactive barrier, as well as how they influence water infiltration and retention. Both approaches created localized moisture variations, which suggests the formation of large biofilm clusters. These clusters occupy a relatively modest fraction of the whole medium, which contributes to explain the resilience of the systems and the observation that they take a long time to clog (incipient clogging was only observed at the end of the continuous recharge operation). The evolution of ERT maps also implies a rather dynamic structure of water flow within the USZ, which preferential flow paths (infiltration fingers) varying in response to changes in the biofilm clusters.

While ERT provides reliable maps of moisture distribution, interpretation is best achieved when coupled to point measurements. In particular, we have found oxygen monitoring particularly useful to add a quantitative measure of biochemical activity to the moisture maps. The combined analysis of hydraulic monitoring, EPS quantification, and ERT imaging demonstrates that recharge strategy and barriers implementation influence biofilm development and water distribution in the USZ. The RBS, due to its organic-rich composition and higher porosity, promotes biofilm formation and retains more moisture, reinforcing a positive feedback loop between microbial activity and hydraulic behavior. ERT imaging proved particularly valuable for visualizing spatial heterogeneity and preferential flow patterns, offering critical insights that complement point-based measurements and support informed design and operation of SAT systems. The findings revealed significant contrasts between the reactive barrier system (RBS) and the control sand system (SandS). The RBS, with enhanced porosity and organic material, retained moisture significantly longer than SandS. This prolonged moisture retention supported sustained biofilm growth, critical for biodegradation and contaminant removal. Conversely, SandS showed faster drainage, which favors faster aeration and, in general, more aerobic conditions.

Cross-hole ERT also identified anomalies and preferential flow patterns that could have an impact on the long-term efficiency of the MAR-SAT systems. For example, under continuous recharge, resistivity data showed increased heterogeneity and the formation of preferential pathways. This highlights the need for adaptive management strategies. ERT's detailed monitoring capabilities are an essential tool for understanding and improving MAR-SAT systems. It enables precise control of critical processes such as biofilm formation and contaminant degradation. Recharge management significantly affects system performance. Pulsed recharge initially facilitated higher biofilm growth rates by enhancing periodic oxygenation and rapid infiltration. In contrast, continuous recharge facilitates steady-state conditions, sustaining consistent biofilm activity over extended periods. These findings emphasize the importance of tailoring recharge strategies to specific site conditions and target contaminants.

Organic matter derived from plants contributes secondary porosity, which retains water for extended periods, supporting biofilm growth and activity during dry cycles. Consequently, systems that maintain higher moisture levels can sustain





contaminant degradation for longer durations post-recharge. During wet cycles, surface water infiltrates through preferential
flow paths, subsequently hydrating the broader unsaturated zone. Similarly, during dry cycles, drainage follows these
heterogeneous pathways.

**Acknowledgments:** This work was supported by Water JPI MARADENTRO (PCI2019-103603 and PCI2019-103425),
CONMIMO (TED2021-131188B-C31), AGAUR-SGR00609, the Spanish Ministry of Science, Innovation and Universities
(MICIU), the AEI, and the European Union NextGenerationEU/PRTR through the program RyC2023 MICIU/AEI
/10.13039/501100011033, and to the EU Life Programme (LIFE REMAR LIFE20 ENV/ES/000284). We extend our gratitude
to the Consorci d'Aigües de la Costa Brava Girona (CACBGi) for their support at the WWTP.

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
