# Peer review of "Mapping Water Content Dynamics in MAR-SAT systems using 3D Electrical Tomography"

_EGUsphere, 2025_

## Author Comment (AC1)

We respond in the spirit of HESSD to promote discussion. Therefore, this document is not meant as a typical "reponse to comments" in which we indicate how we address comments (we will wait for the remaining reviewers). Instead, we discuss the issues raised by Alex Furman in the hope of raing discussion (and, indeed, some of the issues he brings deserve discussion by the soil aquifer treatment (SAT) community.

First of all, we thank Alex Furman for the overall positive assessment of our work and for the constructive tone of the issues he criticizes.

Regarding the, indeed pioneering, work of Haaken et al., you are absolutely right. We believe that SAT systems should be monitored continuously to help during operation and beyond, its pioneering nature, the fact that they buried their electrodes was a very good idea for large scale, long-term, operation.

Regarding Shafdan, it is indeed a paradigmatic example. We use it (and teach it in class!) as an example on what all cities in water scarce regions should do. However, it may not be so relevant for the goals of this paper, centered on the use of ERT. Depending on the comments of other reviewers, we will probably expand the introduction and discussion and make more explicit mention of Shafdan.

Below we address the specific comments in blue our replies.

1. While not the core of this study, it is about time that researchers stop claiming that adsorption is a key element in SAT. A simple mass balance would show that all the adsorption capacity is used at a relatively early stage of the SAT facility life, and additional adsorption is possible only if there exists a degradation process that 'empties' adsorption sites

   Actually, we do not quite agree. While it is true that sorption does not affect the flux of (inorganic) solutes after a transition time, it is fundamental for partitioning compounds (the ones that are toxic by accumulation in human tissues by processes similar to those that control absorption into biofilms). In fact, you hint it at the of your comment ("if there exists a degradation process"). The relevance of sorption lies in the fact that it increases the residence time of contaminants in the microbiologically active zones, thereby favoring biodegradation. In this sense, biofilms represent an important compartment where both sorption and degradation occur in close interaction (Jou-Claus et al., 2024).

2. L68: Biofilms are not the critical element. Microbes are. Biofilms are the environment that helps microbes perform their functions

   We appreciate your clarification regarding the distinction between microbes and biofilms. We fully agree that microbial activity is the key driver of contaminant transformation in SAT systems. However, in practice, the majority of microbes in these systems are embedded within biofilms. This structural organization creates conditions that are not present for planktonic microorganisms: biofilms provide a matrix with sorption capacity (especially relevant for compounds with high logKow), localized redox gradients, and extracellular enzymes that together enhance biodegradation (Jou-Claus et al., 2024 (https://doi.org/10.1021/acs.est.3c08465); Wang et al., 2022

(https://doi.org/10.1016/j.advwatres.2022.104286),2024
(https://doi.org/10.1029/2023WR036872)).

In addition, biofilms are not only relevant for microbial functioning but also represent the main factor responsible for biological clogging and water (and solutes) retention, which has direct implications for the hydraulic performance of SAT systems. For these reasons, we consider biofilms a critical parameter to be managed, as they integrate both the microbial activity and the hydrodynamic constraints of the system.

Apparently, the authors did not read Haaken et al. thoroughly…

R.3- We thank you for pointing this out. Indeed, Haaken et al. conducted pioneering work applying time-lapse ERT to monitor unsaturated zone processes during SAT, and we acknowledge that our current phrasing ("remains unexplored") is misleading. Our intention was to emphasize that, although Haaken et al. demonstrated the feasibility of using ERT in the USZ, the technique has rarely been applied, and its potential for systematically linking geophysical observations with hydro biogeochemical processes in SAT remains underexplored. We wonder if ERT is used to help management at Shafdan.

We will revise the sentence accordingly to properly cite Haaken et al. and clarify the novelty of our work.

3. Interestingly, the authors practically ignore the vast body of literature that comes from the Israeli SAT experience – the largest of its kind that has operated for about 30 years, and is documented quite well in the literature

We are thoroughly familiar with the work at Shafdan, which we consider a paradigm of what all cities in water scarce countries should do. However, the main objective of our manuscript is not to provide a review of SAT systems worldwide (let us wait and see what the other reviewers say). This is why your correction about the work of Haaken et al. is so important. Our goal in this paper is to propose the application of ERT for monitoring infiltration dynamics and linking them to biofilm development in the unsaturated zone. For this reason, we have not included detailed discussion of other large-scale MAR projects. Instead, we focus on demonstrating how ERT can capture hydrological changes associated with biofilm formation, which we consider the critical element affecting system performance.

4. Pulse (it is fine to use a short version, but be consistent and use short or full versions synchronized)

R.5 – We will correct and be consistent.

5. ~L145. Isn't the difference in environmental conditions (temperature) a concern?

Indeed, temperature can influence microbial and geochemical processes in SAT systems and, thus, biofilm growth. Temperature also affects viscosity and oxygen solubility. In our case, both recharge episodes were conducted with relatively stable temperature conditions (Figure S.1). However, the average temperature during the Pulsed WP was slightly higher than during the Continuous WP, with

differences around 7-10 °C. We therefore consider that temperature variations may have had some influence on the results. However, the impact on viscosity is moderate (some 20%) and the impact on microbial conditions is unclear (we have found that, after several years, microbial communities become quite resilient and adapt to temperature changes. This is why we believe that the operational conditions tested are more important. Nevertheless, you are right that the issue cannot be ignored and, depending on the comments by other reviewers, we will add a sentence in the discussion to acknowledge that of course temperature is an environmental factor that may modulate removal processes in SAT systems.

6. Section 2.4. Any petrophysical model to relate resistivity with water content?

   In our case, the main reasons for not implementing a petrophysical model to determine the water content from the obtained electrical resistivity was the high heterogeneity observed in the electrical resistivity 3D distribution due to preferential vertical flow paths. This heterogeneity, caused by the complex infiltration patterns and biofilm development, made it challenging to apply traditional petrophysical relationships that assume more homogeneous conditions. Instead, we prioritized understanding the spatial and temporal patterns of water infiltration and biofilm development rather than obtaining precise quantitative water content values.

7. Is a quarter of a second enough to get charge stability for the reactive layer?

   Our choice of a 250 ms current pulse proved sufficient to achieve charge stability in the reactive layer, as confirmed by our quality-control criterion. By comparing normal and reciprocal measurements and retaining only those pairs with ≤ 15% difference, we ensured that any residual polarization or transient charging had fully equilibrated within each 0.25 s injection. In practice, few measurements were discarded by this threshold.

8. 2-3. These plots are helpful, but zooming in to show a single representative day or two at different periods would be much more interesting

   We thank you for the suggestion. The purpose of Figures 2 and 3 was to illustrate the overall inflow dynamics across the two recharge episodes, and how these controlled soil moisture and water table depth. To help readers also visualize short-term variations, we have already adjusted the scale in the first and last parts of the plots to highlight daily dynamics at the beginning and end of the episodes. Following the reviewer's comment, we will include an additional zoomed-in view of one representative day from each recharge mode (Continuous and Pulsed) in the Supplementary Material.

9. Figures 5-7 are nice, but difficult to analyze. The authors are encouraged to use some integrative type of analysis that would smooth the noise

   You are right! We gave a lot of thought to the presentation and discussion of results. Actually, we hope that readers with available time (hopefully PhD students) will have the time to analyze them in detail and enjoy them as much as we did. In fact, we can conjecture some geostatistical integrative analyses (i.e., evolution of connectivity, cluster sizes, etc.). However, we do not consider that the

technique is sufficiently mature for such analysis. In fact, as you correctly point out in subsequent comments, some of our analyses are somewhat speculative.

10. 5through 7.

R.11- Thanks, we will correct it.

11. L423-426. This is not much more than speculation. Any evidence?

We wrote "The fact that oxygen was virtually depleted at 35 cm during the Puls WC suggests a very high level of biological activity. Perhaps more significant is that Oxygen never reached air saturation, even though it is clear that it was being sucked (presumably across preferential air paths, in a process similar to the "aquifer breathing reported by Roumelis et al. (2025), except that in their case it was driven by aquifer head fluctuations, while ours was driven by drainage fluctuations."

Sure, there is some speculation here. We cannot directly prove the mechanism (hence "suggests"). However, the fact that observed oxygen dynamics are consistent with processes previously described as "aquifer breathing" by Roumeli et al. (2025). In our case, we hypothesize that similar preferential air pathways may exist, though driven by drainage fluctuations rather than aquifer head variations. This is consistent with ERT observation, which is the point we are trying to make. ERT only measures EC, but its time evolution yields rich food for thought.

Still, depending on the other reviewers, we will revise the text to something like: "The fact that oxygen was virtually depleted at 35 cm during the Puls WC suggests intense microbial activity. Interestingly, oxygen never reached full air saturation, despite clear evidence of air ingress. While we cannot directly demonstrate the mechanism, this pattern is consistent with the 'aquifer breathing' process described by Roumelis et al. (2025). In our case, we hypothesize that the pulsed recharge strategy enhanced oxygen entry, thereby promoting aerobic conditions in the unsaturated zone.

12. This is an interesting idea, but it requires much better support. Resistivity 'hot-spots' indicate water content. From here to biochemical hot-spots, the distance is quite far. At best, it indicates the formation of biofilm, but there are so many things that can trigger biofilm formation (some of which are related to microbial stress, not to microbial prosperity

We are not sure which text you are referring to. We believe you refer to "Beyond its potential use for detailed monitoring, ERT's ability to map water distribution heterogeneity has proven useful in gaining insight on the functioning of different recharge strategies and the use of a reactive barrier, as well as how they influence water infiltration and retention. Both approaches created localized moisture variations, which suggests the formation of large biofilm clusters".

We agree with you in that ERT detects electrical conductivity contrasts, which relate directly to water content and not biochemical activity per se. However, biofilms are composed of up to ~97% water and their extracellular polymeric substances (EPS) markedly influence local moisture distribution and hydraulic connectivity. Consequently, the presence of biofilms can enhance water retention

and create localized resistivity anomalies. Our interpretation is therefore not that ERT directly images biofilms, but rather that ERT can provide indirect evidence of their dynamics.

13. L456 and on. Well, understanding that oxygen monitoring is useful is interesting, but given that this is the only thing that was monitored, it puts it in a different light. What about ORP (just an example) – won't that be equally supportive? While I do not argue, it isn't easy to conclude here

Well, not quite true, we monitored T, water content, EPS, EC, etc. We wrote "While ERT provides reliable maps of moisture distribution, interpretation is best achieved when coupled to point measurements. In particular, we have found oxygen monitoring particularly useful to add a quantitative measure of biochemical activity to the moisture maps. The combined analysis of hydraulic monitoring, EPS quantification, and ERT imaging demonstrates that recharge strategy and barriers implementation influence biofilm development and water distribution in the USZ."- Indeed, parameters such as ORP could also provide valuable information as indicators of microbial activity, particularly if measured continuously and non-invasively. However, in this study oxygen was the parameter available for continuous monitoring, and we found it to be particularly useful for assess microbial processes. Our conclusion is therefore based on oxygen dynamics, while we recognize that integrating additional parameters (e.g., ORP, $CO_2$, redox-sensitive solutes) in future work would further strengthen the interpretation.

14. The use of biological materials to enhance biochemical activity is not new, but interesting in the context of ERT. Nevertheless, this practice needs to be discussed in terms of long-term activity. It is not trivial to refresh such a layer after its functionality diminishes

Indeed, in our observations, the reactive barrier primarily serves to accelerate the establishment of a microbial community that is more resilient to environmental changes. Once this community is established, the reactive layer maintains its functionality without the need for periodic renewal, as the consumed organic matter is, at least in part, supplemented by root growth. Still, as you know from Shafdan, the long time behavior is somewhat uncertain. We have only been observing for 4 years. Still, we have given some thought on the refreshing of the layer (note that its behavior improves over time!), so certainly we do not want to build it anew.

You mention past use of biological materials. We are not aware of this.

---

## Author Comment (AC2)

SAT is a form of MAR. Discussion on the comments provided by Prof. Xinqiang Du.

Lurdes Martinez-Landa[1,2, *], Juan José Ledo[3], Perla Piña-Varas[4], Jesús Carrera[2,5], Paola Sepúlveda-Ruiz[6], Montserrat Folch[6], Cristina Valhondo[2,5, *]

[1] Department of Civil and Environmental Engineering, Universitat Politècnica de Catalunya, Barcelona, Spain
[2] Associated Unit: Hydrogeology Group (UPC–CSIC)
[3] Department of Earth Physics and Astrophysics, Faculty of Physics, Universidad Complutense de Madrid, Spain
[4] Geomodels-UB Research Institute, Faculty of Earth Sciences, Universitat de Barcelona, Spain
[5] Geosciences department, Institute of Environmental Assessment and Water Research, Spanish Research Council (IDAEA-CSIC), Barcelona, Spain
[6] Biology, Sanitation and Environmental Department, University of Barcelona, Av. Joan XXIII, 08028 Barcelona, Spain

We provide this reply in the spirit of HESSD, aiming to foster constructive scientific discussion. For this reason, the present document should not be interpreted as a conventional "response to reviewers," where each comment is accompanied by a detailed revision note (we will do so once all reviewers' reports are available). Instead, our intention is to reflect on and engage with the points raised by Prof. Xinqiang Du, whom we thank for his comments and the opportunity to further discuss the important issues of Managed Aquifer Recharge (MAR) and Soil-Aquifer Treatment (SAT).

Prior to a detailed discussion of the specific points, we wish to address two issues Prof. Xinqiang Du raises regarding (1) whether SAT can be considered a MAR technique and (2) the when scientific references are needed.

Regarding the appropriateness of referring to SAT as a MAR technique, we respectfully argue that SAT is, and should be, considered a subset of MAR. We argue this point, both from a logical perspective, and from its wide use by both the scientific literature and the water management sector. While it is true that SAT's primary focus is on water quality improvement, the process inherently involves the intentional inflow (recharge) of water into an aquifer, thereby also contributing to groundwater resource augmentation, the core purpose of MAR. This inclusion of SAT under the MAR umbrella is well established in foundational and contemporary literature (e.g., Bouwer, 1991, 2002; Dillon, 2005; Bekele et al., 2011; Alharbi and El-Rawy, 2024, among others), but also in the professional sector, perhaps best summarized in the paper prepared by the IAH group on MAR (Dillon et al., 2019).

Regarding scientific referencing, we believe that referencing plays three roles: (1) providing support to a statement (e.g., we believe that SAT is a MAR technique because Dillon (2005) said so); (2) acknowledging the author of an idea (e.g., Bouwer, 1991); and/or (3) providing the reader with a source of additional details. These principles are not universally accepted, but they are the ones we try to follow. Note that, according to these principles, we argue that references are not needed for (1) concepts presented in the paper; (2) concepts that are widely known and accepted by the scientific community of the journal (e.g., we do not provide a reference for Darcy's Law because all HSS readers are familiar with it); and (3) trivial concepts.

Below we address the specific comments in blue our replies.

CC2-1- From an academic classification perspective, Soil-Aquifer Treatment (SAT) is not universally recognized as a subset of Managed Aquifer Recharge (MAR). According to standard definitions, MAR refers to the intentional recharge of water to aquifers for subsequent use or environmental benefit. At the same time, SAT primarily focuses on improving water quality through soil infiltration. Consequently, the term "MAR-SAT" does not represent a commonly accepted implementation scenario in the literature. Given the study's core focus on unsaturated zone monitoring via ERT, the standalone term "SAT" is sufficiently precise to contextualize the research. The explicit association with MAR is unnecessary unless the authors can demonstrate direct relevance to MAR's core objectives.

RCC2-1- We agree that the present study was conducted within a specific SAT system. Our initial choice to include the term "MAR" was motivated by the fact that the methodological approach we present—ERT monitoring of water movement through the unsaturated zone—is directly transferable to a broad range of Managed Aquifer Recharge (MAR) settings. In many MAR implementations, infiltration through the vadose zone plays a central role, and understanding the dynamics of percolation and water–soil interactions is equally relevant.

However, we acknowledge that SAT constitutes a well-defined category on its own and that using "SAT" alone is sufficiently precise for describing the experimental context of this work. To avoid ambiguity in terminology, we have therefore removed the explicit reference to MAR in the title. The revised title now reads: **"Mapping Water Content Dynamics in SAT Systems Using 3D Electrical Tomography."**

As argued in the introduction to this rely, we consider SAT as a subset of MAR.

CC2-2- Line 46-50, the natural pollutant reduction capacity of the porous media can be seen as an additional guarantee to MAR. So, the mentioned regulations are not a constraint for widespread application of MAR. Although the citation supports the conclusion, it is not a reasonable opinion.

RCC2-2- We agree with Prof. Xinqiang Du that the natural attenuation capacity of the soil and sub-surface is an additional guarantee in many MAR applications. Our intention was not to claim that regulations universally hinder MAR implementation, but to highlight those regulatory frameworks fail to acknowledge water quality improvement processes during soil passage.

When the same source-water quality requirements are imposed on all MAR types, the potential of infiltration-based systems to act as effective nature-based solutions for water quality improvement becomes hindered (actually, some of us argue that they effectively favor indirect pollution of groundwater, but it would be long to argue here). Techniques such as infiltration ponds, percolation tanks, and SAT systems rely on the unsaturated zone for filtration, sorption, and biodegradation to remove pathogens, organic contaminants, and nutrients, a geo-purification capacity widely acknowledged in the literature (e.g., Dillon, 2005; Bekele et al., 2011). However, in many countries, the legal framework for aquifer recharge is so restrictive that the feasibility of implementing SAT for water renaturalization at an industrial scale is severely limited. In countries such as France, Spain, and Chile, the use of treated wastewater for recharge is often permitted only at a pilot scale and for research purposes, requiring specific exemptions (Casanova et al., 2016., Miquel, 2003; Rivera-Vidal et al., 2025). This regulatory reality effectively constrains the widespread application of SAT systems, regardless of their proven geo-purification potential (rainfall fails to meet the requirements because of its low pH and the fact that samples usually contain suspended solids).

Thus, our statement aimed to reflect that, in practice, overly stringent or non-differentiated source-water requirements may limit the adoption of infiltration-based MAR systems specifically designed to take advantage of natural attenuation in the unsaturated zone, even though such attenuation indeed represents an added level of protection.

CC2-3- Line 51, "…… for Soil Aquifer Treatment (SAT), a MAR technique that uses treated wastewater as a source for recharge" is a misleading interpretation of SAT and MAR. The reason is the same as in the above comment.

RCC2-3- We acknowledge the perspective of Prof. Xinqiang Du on this point, which mirrors the previous comment on the SAT-MAR relationship. However, we respectfully maintain that SAT is widely considered a form of Managed Aquifer Recharge (MAR) in both the scientific literature and the water management sector, because it necessarily involves the intentional recharge of treated water into an aquifer, thereby increasing groundwater storage as part of the treatment process.

CC2-4- Line 51–55,   Treated wastewater is not a typical recharge source for MAR in most regions, and SAT is not widely recommended as a water quality improvement technology for MAR systems. This is primarily because pollution prevention and contaminant attenuation are paramount considerations for MAR, and SAT may not consistently meet the stringent water quality requirements for aquifer recharge without additional validation.

RCC2-4- We agree that in many regions treated wastewater is not yet a common source for MAR implementation, largely due to stringent regulatory requirements and the need to ensure the protection of groundwater resources. We also acknowledge that SAT is not universally recommended as a water-quality improvement step within MAR schemes unless performance has been demonstrated under site-specific conditions.

However, our intention in this section was not to imply that SAT is widely deployed for MAR using treated wastewater, but rather to summarize the established evidence showing that SAT can provide substantial water-quality improvements when such applications are permitted. Numerous studies have documented the ability of SAT to remove suspended solids, pathogens, nutrients, dissolved organic carbon, and a range of organic and inorganic contaminants (e.g. Valhondo et al., 2020, Sanz et al., 2024, 2024b).

CC2-5- Line 114–115,   The statement "following the recommendations of a soil scientist" does not constitute a for the described methodology. To justify the approach's rationality, the authors must cite relevant peer-reviewed literature or standardized protocols that support the adopted procedures.

RCC2-5- Indeed, Prof. Xinqiang Du is right, it is not a rigorous scientific basis, it is just a practical recommendation. We delete "following the recommendations of a soil scientist", which still describes the practical solution we used (we hope others can use it).

CC2-6- Line 144–145,   Two critical concerns arise regarding clogging-related claims:

- Logical Inconsistency: Clogging is defined as the phenomenon of reduced hydraulic conductivity. The absence of operational shutdowns for maintenance does not equate to the absence of clogging—mild to moderate clogging can occur without necessitating downtime. Notably, the statement in Line 224 ("we have observed the development of a water layer due to the reduction of infiltration capacity") directly

indicates clogging-induced reduced permeability, creating a contradiction that requires resolution.

- Insufficient Evidence: The claim regarding vegetation's role in clogging control lacks empirical support. The authors must provide verified references demonstrating vegetation's efficacy in mitigating clogging in SAT/MAR systems, or supplement with in-situ data (e.g., vegetation root distribution, clogging layer composition).

RCC2-6- With some qualifications, we agree on both accounts:

First, regarding clogging, it is true that the absence of operational shutdowns does not imply that clogging does not occur (and we will reword the statement in the revised version of the paper, if accepted). What we have observed is that, in some cases, continuous operation helps sustain recharge for longer periods. The presence of a water layer on the basin surface does not necessarily indicate a reduction in infiltration capacity if this layer remains stable over time. The issue arises when the water layer increases abruptly, which suggests a sudden accumulation of fine materials that reduce infiltration capacity, possibly due to an overload at the WWTP outlet causing the mobilization of fines.

Second, regarding the role of plants, while the statement is marginal to the goal of the paper, the issue is not well known in SAT literature. Therefore, in the revised version, we will provide references both from our own experience in SAT and from the forest literature (e.g., Valhondo et al., 2020., Wu et al., 2017., Le Coustumer et al., 2012). We have observed in pilot basins of 400 and 5,000 m² that vegetation contributes to stabilizing the free movement of organic particles, prevent excessive algal growth by covering the water surface by reducing sunlight exposure, and limits the mobilization of fines from the basin slopes, thereby preventing their accumulation across the entire recharge basin surface. Superficial clogging was no longer observed once the plants covered the basin surface, under the same recharge scheme. Specifically, in the 5,000 m² basin, the complete removal of vegetation resulted in significant clogging problems in this system.

In fact, we are preparing a manuscript presenting the results in the 400 m$^2$ pilot basins, in which we discuss these observations.

CC2-7- Line 165, Formatting error: "250 10$^{-3}$" should be corrected to "250×10$^{-3}$".

RCC2-7- Thanks, we have corrected it

CC2-8- Line 206, Terminology Ambiguity: The phrase "aquifer head (depth to water)" is imprecise.

RCC2-8- Thank you, indeed, we will change it to "depth to the water table" in the revised manuscript.

CC2-9- Line 393–395, The opinions in this section lack support from validated peer-reviewed references. To enhance credibility, the authors must supplement with relevant literature that corroborates the proposed mechanisms or conclusions.

RCC2-9- This interpretation is based on direct, site-specific evidence. The recorded EC peaks show a precise temporal correlation with independent meteorological data on coastal windstorms, which are the known driver of seawater overtopping events in this area. Furthermore, the WWTP operators consistently document salinity surges in the influent

following such storms, confirming that seawater intrusion into the coastal sewerage network is the established cause. Thus, this is not a proposed mechanism but a documented occurrence for this specific system.

We feel shy to consider this a contribution of our paper, but rather a frequent observation. Further, this statement is an explanation of the supporting information, which is not relevant to the paper. Depending on what other reviewers say, we will either delete the statement or expand the explanation in the SI of the revised version, at our site and other coastal cities, because it may be relevant for the operation of WWTPs and SAT systems. Yet, to our knowledge, it has not been addressed in the scientific literature.

CC2-10- Line 442-448, I think the conclusion that ERT allows visualizing and monitoring variations in water retention and biofilm growth is a verified fact in the past relevant research. So, the authors should keep up with the latest developments.

RCC2-10- Definitely, numerous authors, including ourselves, have used ERT for monitoring variations in water retention of soils. We do cite six such papers (there are many others). There are also some, far less, that have used ERT for SAT. And those have centered on characterization purposes (e.g., Sendrós et al., 2020) or monitoring water flow (e.g. Haaken et al., 2016, or Arboleda-Zapata et al, 2025). If the paper is accepted, we will cite these papers

CC2-11- A critical gap in the manuscript is the lack of discussion on how the study's conclusions translate to MAR applications. Specifically: What actionable principles or technical guidelines can be derived to inform MAR system design, operation, or performance assessment? How does the ERT-based monitoring approach address key challenges in MAR (e.g., recharge rate optimization, contamination risk mitigation)? The authors should supplement this section with concrete implications for MAR practice to enhance the study's applied value.

RCC2-11- We thank Prof. Du for this comment. We will add a closing remark to emphasize the importance of monitoring for the optimal operation of the system. Note, however, that optimal operation does not necessarily address the large questions posed by Prof. Du. Regarding recharge rate, SAT systems are usually designed for a given flow rate. Thus, our comment will mention the possibility of seasonal variations in flow rate depending on ERT response. As for contaminant mitigation, ERT has helped us understand the dynamism of biofilms. While this contributes to explain the outstanding behavior of SAT in contaminant removal, we do not think our understanding is sufficiently mature to to link ERT response to contaminant removal. Still, we will mention it as a future challenge.

References:

Alharbi, H., El-Rawy, M. (2024). Soil Aquifer Treatment (SAT) for Managed Aquifer Recharge and Water Quality Improvement in the MENA Region. In: El-Rawy, M., Negm, A. (eds) Managed Aquifer Recharge in MENA Countries. Earth and Environmental Sciences Library. Springer, Cham. https://doi.org/10.1007/978-3-031-58764-1_4

Arboleda-Zapata, M., Osterman, G., Li, X., Sasidharan, S., Dahlke, H. E., & Bradford, S. A. (2025). Time-lapse ensemble-based electrical resistivity tomography to monitor water flow from managed aquifer recharge operations. Journal of Hydrology, 659, 133282.

Bekele, E., Toze, S., Patterson, B., / Higginson, S. Managed aquifer recharge of treated wastewater: Water quality changes resulting from infiltration through the vadose zone. Water

Research, 45, 5764-5772 (2011). http://www.sciencedirect.com/science/article/pii/S0043135411005100

Bouwer, H. Artificial recharge of groundwater: hydrogeology and engineering. *Hydrogeology Journal* **10**, 121–142 (2002). https://doi.org/10.1007/s10040-001-0182-4

Bouwer, H. Role of Groundwater Recharge in Treatment and Storage of Wastewater for Reuse. Water Science and Technology 24, 295-302 (1991). https://doi.org/10.2166/wst.1991.0258

Casanova, J., Devau, N., Pettenati, M. (2016). Managed Aquifer Recharge: An Overview of Issues and Options. In: Jakeman, A.J., Barreteau, O., Hunt, R.J., Rinaudo, JD., Ross, A. (eds) Integrated Groundwater Management. Springer, Cham. https://doi.org/10.1007/978-3-319-23576-9_16

Dillon, P. Future management of aquifer recharge. Hydrogeol J 13, 313–316 (2005). https://doi.org/10.1007/s10040-004-0413-6

Dillon, P., Stuyfzand, P., Grischek, T., Lluria, M., Pyne, R. D. G., Jain, R. C., ... & Sapiano, M. (2019). Sixty years of global progress in managed aquifer recharge. Hydrogeology journal, 27(1), 1-30.

Haaken, K., Furman, A., Weisbrod, N., & Kemna, A. (2016). Time-lapse electrical imaging of water infiltration in the context of soil aquifer treatment. *Vadose Zone Journal*, *15*(11), 1-12.

Le Coustumer, S.L., Fletcher, T.D., Deletic, A., Barraud, S., Poelsma, P., 2012. The influence of design parameters on clogging of stormwater biofilters: a large-scale column study. Water Res. 46, 6743e6752. https://doi.org/10.1016/j.watres.2012.01.026

Miquel G (2003) La qualité de l'eau et de l'assainissement en France. Rapport de l'Office parlementaire d'évaluation des choix scientifiques et technologiques, Sénat n°215, Assemblée Nationale n°705, 2 tomes.

Rivera-Vidal, R., Aruní, J. A., Melo, O., Delgado, V., Parra, V., Stehr, A., and Daniele, L. Managed aquifer recharge implementation challenges: Lessons from Chile's water-scarce regions. Groundwater for Sustainable Development, 31. 2025. https://doi.org/10.1016/j.gsd.2025.101502

Sendrós, A., Himi, M., Lovera, R., Rivero, L., Garcia-Artigas, R., Urruela, A., & Casas, A. (2020). Geophysical characterization of hydraulic properties around a managed aquifer recharge system over the Llobregat River Alluvial Aquifer (Barcelona Metropolitan Area). Water, 12(12), 3455.

Sanz, C., Sunyer-Caldú, A., Casado, M., Mansilla, S., Martinez-Landa, L., Valhondo, C., Gil-Solsona, R., Gago-Ferrero, P., Portugal, J., Diaz-Cruz, M. S. Carrera, J., Piña, B., Navarro-Martín, L. Efficient removal of toxicity associated to wastewater treatment plant effluents by enhanced Soil Aquifer Treatment. Journal of Hazardous Materials, Vol. 465 (2024). https://doi.org/10.1016/j.jhazmat.2023.133377

Sanz, C., Casado, M., Martinez-Landa, L., Valhondo, C., Amalfitano, S., Di Pippo, F., Levantesi, C., Carrera, J., Piña, B. Efficient removal of antibiotic resistance genes and of enteric bacteria from reclaimed wastewater by enhanced Soil Aquifer Treatments. Science of The Total Environment, Vol. 953 (2024). https://doi.org/10.1016/j.scitotenv.2024.176078

Valhondo, Cristina., Martínez-Landa, Lurdes., Carrera, Jesús., Díaz-Cruz, Silvia M., Amalfitano, Stefano., Levantesi, Caterina. Six artificial recharge pilot replicates to gain insight into water quality enhancement processes. Chemosphere, 240 (2020). https://doi.org/10.1016/j.chemosphere.2019.124826

Wu, G.-L., Liu, Y., Yang, Z., Cui, Z., Deng, L., Chang, X.-F., Shi, Z.-H., 2017. Root channels to indicate the increase in soil matrix water infiltration capacity of arid reclaimed mine soils. J. Hydrol. 546, 133e139. https://doi.org/10.1016/ j.jhydrol.2016.12.047.

---

## Author Comment (AC3)

We sincerely thank Referee #2 for the careful reading of the manuscript, the positive assessment of its scientific contribution, and the constructive suggestions provided (https://doi.org/10.5194/egusphere-2025-3994-RC2). The reviewer's comments are highly appreciated and will help improve the readability, technical accuracy, and overall quality of the paper.

Below, we address the specific comments raised by the referee, with our responses provided in blue.

**R2.1**-This paper applies a comprehensive set of techniques to monitor the performance of a pilot-scale sustainable water-treatment system, characterised by low energy demand and reduced chemical consumption compared with conventional alternatives. It provides a clear, detailed, and concise description of the methodology and pilot-plant setup, along with a thorough analysis of the results, effectively leveraging the information obtained from each monitored variable. Moreover, I consider that the study represents a clearly written step forward in understanding how the choice of infiltration media and the recharge strategy—pulsed versus continuous—affect system behaviour, water content, and biofilm development within the unsaturated zone. The work also offers a solid and well-framed discussion of the results, integrating recent and relevant literature to contextualise the findings and highlight their contribution to current knowledge.

**R2.1**- We sincerely thank the referee for this very positive and encouraging assessment of our work. We greatly appreciate the recognition of the comprehensive monitoring approach, the clarity of the methodological description, and the relevance of the discussion in advancing the understanding of SAT system behavior. These comments are highly motivating and confirm the relevance of the study within the context of sustainable MAR and SAT research.

Please find below minor comments to improve the reading and understanding of your manuscript:

**R2.2**-Line 202: Please consider changing "Supporting Information" to "Supplementary Material"

**R2.2**- Thank you for this suggestion. We have renamed "Supporting Information" as "Supplementary Material" throughout the manuscript (Line 202 and 117)

**R2.3**-Line 228: Please remove ".." at the end of the line.

**R2.3**- Yes, it has been removed.

**R2.4**-Line 241: I would recommend reorganising the figures in the Supplementary Material so that their numbering follows the order in which they are quoted in the manuscript. Currently, Figures SI.10 and SI.11 appear before SI.8 and SI.9 in the text.

**R2.4**- Thank you for pointing this out. We have reorganized the Supplementary Material so that the figures appear in the order in which they are cited in the manuscript.

**R2.5**-Line 298: Please double-check "ER". It seems this should be "ERT".

**R2.5**- Indeed, thank you for noticing this. The typo has been corrected to "ERT".

**R2.6-**Line 364: Change "Figure 5(" by "Figure 5 ("

**R2.6**- Changed, thanks!

**R2.7-**Line 378: Please consider adding the method for determining the permeability value of 10 m/day. I consider it would be beneficial for the readers to have this information.

**R2.7**- We have obtained this approximation using the travel time of the EC peaks produced by the sea water entrance into the sewer system during some storms with specific wind orientation. These picks are easily identified and measured in the recharge basin surface and in the O piezometer located in the USZ base, acting as a tracer test.

Supplementary material:

**R2.8-** Consider moving the explanation of Figure SI. 3 before its first mention in the text.

**R2.8**- Thank you for the suggestion but the figures in the Supplementary Material follow a numerical order from Figures S1 to S7, consistent in both documents. For Figures S8 to S11, the order of Sections S2.4 and S2.5 sections in the Supplementary Material has been rearranged to follow the order in which they are cited in the main text.

**R2.9-** Double checked the use of "v" in "Aquacheck prove. The Aquacheck probe

**R2.9**- Yes, Thanks!